# Plexins as Regulators of Cancer Cell Proliferation, Migration, and Invasivity

**DOI:** 10.3390/cancers15164046

**Published:** 2023-08-10

**Authors:** Shira Toledano, Gera Neufeld

**Affiliations:** The Cancer Research Center, The Bruce Rappaport Faculty of Medicine, Technion-Israel Institute of Technology, Haifa 3109602, Israel; sshirato@campus.technion.ac.il

**Keywords:** semaphorins, cancer, plexins, metastasis

## Abstract

**Simple Summary:**

Cancer is the second leading cause of death in the US. In 2023, approximately 609,820 deaths from cancer are expected. Plexins are a family of receptors that directly affect tumor cell invasiveness and proliferation and, as a result, promote or inhibit tumor progression and, as such, may also serve as potential biomarkers for various types of cancer. The main ligands of the plexin family of receptors are semaphorins. Here, we review only the direct effects of plexins-mediated signal transduction on the behavior of various types of tumor cells, excluding their effects on the various other types of cells that are recruited to the tumor microenvironment. Plexins also transduce the signals of additional ligands and associate with various cell membrane-bound and intracellular modulators that, in turn, modify their effects.

**Abstract:**

Plexins are a family of nine single-pass transmembrane receptors with a conserved GTPase activating protein (GAP) domain. The plexin family is divided into four subfamilies: Type-A, type-B, type-C, and type-D plexins. Plexins function as receptors for axon guidance factors of the semaphorin family. The semaphorin gene family contains 22 genes that are divided into eight subclasses of which subclasses three to seven represent vertebrate semaphorins. The plexins and their semaphorin ligands have important roles as regulators of angiogenesis, cancer proliferation, and metastasis. Class 3 semaphorins, with the exception of sema3E, are the only semaphorins that do not bind directly to plexins. In order to transduce their signals, they bind instead to complexes consisting of receptors of the neuropilin family and various plexins. Some plexins also form complexes with tyrosine-kinase receptors such as the epidermal growth factor receptor ErbB2, the mesenchymal epithelial transition factor receptor (MET), and the Vascular endothelial growth factor receptor 2 (VEGFR2) and, as a result, can modulate cell proliferation and tumor progression. This review focuses on the roles of the different plexins in the control of cancer cell proliferation and invasiveness. Plexins also affect tumor progression and tumor metastasis by indirect mechanisms, such as modulation of angiogenesis and immune responses. However, these topics are not covered in the present review.

## 1. The Plexin Receptor Family

The nine receptors of the plexin family are segregated into four subfamilies consisting of four Type-A plexins, three Type-B plexins, and single C and D plexins (Figure 1) [1]. Plexins serve as direct binding receptors for most semaphorins, which are a large family of evolutionarily conserved signaling molecules that were initially identified as axon guidance factors [2]. The extracellular domains of all plexins contain a sema domain which is also present in semaphorins and serves as an auto-inhibitory domain in the basal, dimeric, non-activated state of the receptor [3]. Once a semaphorin binds to the extracellular domain of the plexins, it induces a conformational change in the two dimerized plexins that initiates signal transduction [3,4,5]. The extracellular domains of the plexins and of the semaphorins also contain plexin-semaphorin-integrin (PSI) domains which contain eight cysteine residues and bridge the sema and immunoglobulin-plexin-transcription (IPT) domains thereby ensuring the correct formation and correct orientation of the ligand-receptor binding sites [1]. The extracellular domain of the plexin-B family receptors is unique among the plexins because it contains a conserved cleavage site for furin-like pro-protein convertases. This cleavage site is posttranslationally processed, resulting in the generation of functional soluble type B plexin extracellular domains that are able to sequester semaphorins that bind to type B plexins [6]. The intracellular domains of the plexins are characterized by the presence of a GTPase-activating (GAP) domain. The GAP domain is highly conserved throughout the plexin family. Activation of plexin signal transduction is associated with the recruitment and activation of several GTP-binding proteins, such as Rac1, Rnd1, and Rho, to the GAP domain [7]. Activation of the GAP domain also confers the deactivation of R-Ras, M-Ras, and Rap1 in all plexins [8,9,10,11]. The GAP activity toward R-Ras and M-Ras, but not toward Rap1, also requires Rnd GTPases binding to the plexin receptors [8,9,11,12]. The intracellular domains also contain putative tyrosine phosphorylation sites and a split cytoplasmic SP (sex-plexin) domain but no tyrosine kinase domain (Figure 1) [13].

Plexin-A1 is the best-studied plexin among the vertebrate type-A plexins. Its cytoplasmatic domain was found to be critical for its ability to mediate sema3A signal transduction [13,14,15]. Following stimulation by sema3A, the Rac1 guanyl nucleotide exchange factor (GEF) FARP2 and Rac1 bind to plexin-A1, resulting in the activation of Rac1, which subsequently promotes WNT3A-induced accumulation of β-catenin in the nucleus [16]. Active Rac1 also binds to the RhoGTPase Binding Domain (RBD) of plexin-A1 and, as a result, induces a conformational change in the intracellular domain of plexA1, which enables the binding of Rnd1 to plexin-A1. Interestingly, RhoD is also able to bind to the RBD domain and can inhibit the binding of Rnd1 [17,18]. Type-A plexins also interact directly with Molecules Interacting with CasL (MICALs). The three MICAL human family members are flavoprotein oxidoreductases that oxidize actin filaments in response to stimulation with semaphorins causing their disassembly [19].

In contrast to type-A plexins, type-B plexins contain a unique C-terminal PDZ binding motif [20]. Through this domain type-B plexins associate with PDZ-RhoGEF and with Leukemia-associated RhoGEF (LARG). Rnd1 promotes this association by binding to plexin-B1 [21]. It was also found that activation of plexin-B1 by sema4D creates docking sites for the SH2 domains of phospholipase C-gamma (PLCgamma). Recruited PLCgamma subsequently then activates PDZ-RhoGEF [22]. The PDZ binding motif is also critical for the activation of the plexin-B family by RhoA, resulting in stress fiber formation [23]. Another regulator of Rho activity is p190 Rho-GTPase, which inactivates Rho upon the binding of semaphorins to plexins [24]. Unlike the type-A and Type-C plexins, plexin-D1 also possesses a PDZ binding motif [25].

Type-A and Type-B plexins associate spontaneously to form homodimers [4,5] or heterodimers [26]. Class 4–7 semaphorins bind directly to plexins. Class-3 semaphorins, with the exception of sema3E which binds directly to plexin-D1, bind to a complex consisting of a plexin receptor, and one of the two neuropilin receptors that on their own do not transduce semaphorin signals [13,27].

Type-B plexins were reported to associate with the tyrosine-kinase receptor MET [28]. Sema4D binds to plexin-B1 and plexin-B2 receptors, and it was observed that when plexin-B1 is associated with the MET receptor and with its close homolog receptor Ron, sema4D induces their auto-phosphorylation and promotes invasiveness of tumor cells expressing these receptors as well as tumor angiogenesis [29,30]. Likewise, it was found that sema3C can also transactivate several tyrosine-kinase receptors, including EGFR, ErbB2, and MET [31]. Plexins can also form complexes with the vascular endothelial growth factor (VEGF) tyrosine-kinase receptor VEGFR2. In the presence of sema6D, plexin-A1 receptors associate with VEGFR2 and promote the proliferation of malignant mesothelioma cells [32]. Additionally, it was discovered that plexin-A4 can also associate with VEGFR2 and with the fibroblast growth factor tyrosine-kinase receptor FGFR1. This, in turn, enhanced the VEGF- and FGF-induced phosphorylation of these tyrosine-kinase receptors [26]. Plexins are also able to associate with the epidermal growth factor receptor ErbB2. ErbB2 can promote pro-migratory and angiogenesis effects by activation and phosphorylation of plexin-B1 and, as a result, activation of RhoA via RhoGEF [22,33]. Additionally, it was reported that sema3E induces invasiveness and metastatic spreading of tumor cells through the transactivation of ErbB2 by a post-translationally modified sema3E that was truncated by furin-like pro-protein convertases (P61-sema3E). In contrast to full-length sema3E, p61-Sema3E induces the association of plexin-D1 with ErbB2 and the phosphorylation of ErbB2 and, as a result, induced tumor cell metastasis [34,35]. Plexins also associate with additional cytoplasmic tyrosine kinases, except from VEGFR2 and ErbB2, such as FES, SRC, PYK2, and FYN [36].

In this review, we focus on the effects that signal transduction mediated by the receptors of the plexin receptor family has on the proliferation, migration, and invasiveness of tumor cells. Signal transduction by plexin receptors also modulates the behavior of non-transformed cells recruited into the tumor microenvironment, such as endothelial cells and various types of immune cells. We do not cover these topics in this review as they are covered elsewhere [37,38,39,40].

## 2. The Role of the Different Plexins in Tumor Progression

### 2.1. Type-A Plexins

Class-3 semaphorins transduce repulsive signals using complexes of neuropilin and type-A plexin receptors. Activation of these plexins by class-3 semaphorins as well as by class-6 semaphorins usually inhibits tumor angiogenesis, and when expressed in tumor cells, these plexins also usually inhibit tumor progression. However, this is a generalization, and their association with additional modulators can lead to opposite effects.

#### 2.1.1. Plexin-A1

Plexin-A1, along with plexin-A4, is a major transducer of sema3A signals when associated with neuropilin-1 (NRP1) [26,41,42]. While plexin-A1 clearly conveys sema3A-induced anti-angiogenic signals [43,44], the effects of plexin-A1 transduced sema3A signals on tumor cells are not as clear. Plexin-A1 was reported to convey sema3A inhibitory signals in malignant mesothelial cells inhibiting their proliferation (Figure 2) (Table 1) [45].

In contrast, it was reported that perlecan interacts with a complex of sema3A, plexin-A1, and NRP-1 on the cell surface of prostate cancer cells and that cleavage of this complex by MMP-7 cell-cell bonds promotes the metastatic dissemination of prostate cancer cells (Table 1) (Figure 3) [46]. Plexin-A1 expression was found to be significantly downregulated in sections from tissues of invasive human breast cancer [47]. It was also reported that plexin-A1 can form a complex with VEGFR2 in malignant pleural mesothelioma cells. Sema6D binds to plexin-A1 and promotes the phosphorylation of VEGFR2. As a result, sema6D activates NF-KB and mediates survival and tumor growth (Table 1) (Figure 3) [32].

**Table 1 cancers-15-04046-t001:** Plexins and their role in various types of cancers. Shown is a summary of the effects of the various plexins and of their various ligands and associated cell surface molecules, on the behavior of various types of tumor cells.

Plexin	Modulators of Plexin-Mediated Signal Transduction	Role	Cancer Type	Refs
**Plexin-A1**	Sema3A	Inhibition of proliferation	Malignant mesothelial cells	[45]
	Sema3A, NRP-1, Perlecan	Promotion of metastatic dissemination	Prostate cancer cells	[46]
	Sema6D, VEGFR-2	Promotion of survival and tumor growth	Malignant mesothelioma cells	[32]
	Sema3A	Promotion of proliferation and glycolytic activity	Lung cancer cells	[48]
	Sema3A, NRP-1	Inhibition of proliferation	Brain tumor stem cells	[49]
**Plexin-A2**		Enhancement of migration and invasion	Prostate cancer cells	[50]
	Sema3C, NRP-1, MAOA, MET	Promotion of perineural invasion	Prostate cancer cells	[51]
	Sema3A, KIAA1199	Inhibition of apoptosis	Cervical cancer cells	[52]
		Enabling cell proliferation and the development of tumors	Glioblastoma derived cells	[53]
**Plexin-A3**		Inhibition of cell invasion	Epithelial ovarian cancer cells	[54]
**Plexin-A4**	Sema6B	Promotion of pro- proliferative signals	Glioblastoma-derived cells, lung-cancer-derived cells, malignant-melanoma-derived cells	[26]
	miR-564	Promotion of cell proliferation and migration	Non-small cell lung carcinoma cells	[55]
**Plexin-B1**	Sema4D	Promotion of EMT and tumor cell metastasis	Head and neck squamous cell carcinoma	[56]
	Sema4D	Promotion of cell invasion, proliferation, and migration	Osteosarcoma cells	[57]
	Sema3C, NRP-1/2, EGFR, ErbB2, MET	Promotion of cancer growth	Castration-resistant prostate cancer cells	[31]
	TMPRSS2-ERG	Promotion of cell migration and invasion	Prostate cancer cells	[58]
		Promotion of cell migration and invasion	Ovarian cancer derived cells	[59]
		Inhibition of breast cancer cell motility	Breast cancer cells	[60]
		Suppression of tumorigenesis	Primary melanoma cells	[61]
		Inhibition of cell proliferation	Basal cell carcinoma cells	[62]
**Plexin-B2**		Inhibition of cell proliferation	Basal cell carcinoma cells	[62]
	Sema4C, ErbB2	Promotion of proliferation and development of tumor metastasis	Breast cancer derived cells	[63]
	Sema4C, MET	Promotion of glioma and glioblastoma cell invasion	Glioma and Glioblastoma cells	[64]
	Angiogenin	Inhibition of tumor cell proliferation and inhibition of tumor development	Glioblastoma cells, breast cancer cells, and myelogenous leukemia cells	[65]
	Angiogenin	Enhancement of CSC stemness and resistance to chemotherapy	Prostate cancer stem cells	[66]
	Angiogenin	Promotion of proliferation, invasion, and tumor growth	Glioblastoma cells	[67]
	Circular RNA, Circ_0013958	Promotion of proliferation, migration, invasion, and tumor growth	Ovarian cancer cells	[68]
	EGFR	Promotion of proliferation, invasiveness and tumor-forming ability by constitutively active plexin-B2	Stem cells from cancers from unknown primary tumors	[69]
**Plexin-B3**	Sema5A	Inhibition of migration and invasion	Glioma cells	[70]
		Inhibition of migration, invasion, and tumor metastasis	Pancreatic cancer cells	[71]
		Promotion of cancer cell growth, cell migration, cell invasion, and tumor progression	Triple-negative breast cancer cells	[72]
**Plexin-C1**	Sema7A	Inhibition of tumor progression	Melanoma cells	[73]
		Inhibition of tumor progression	Glioma cells	[74]
	Sema7A, β1 integrin receptors	Promotion of metastasis	Melanoma and breast cancer cells	[75]
		Promotion of migration and proliferation	Gastric cancer cells	[76]
**Plexin-D1**	P61-sema3E, ErbB2	Promotion of tumor cell invasiveness and tumor cell metastasis	Melanoma cells, lung carcinoma cells, colon carcinoma cells	[34,35]
	NR4A1	Promotion of apoptosis	Breast cancer cells	[77]
	Sema3C, plexin-A2, NRP-1	Promotion of cell survival	Glioma stem cells	[78]

Additionally, although sema3A usually functions as a tumor suppressor, it was discovered that plexin-A1 can mediate sema3A oncogenic signals in lung cancer cells. Inhibition of plexin-A1 decreased cell proliferation in these tumor cells and suppressed their glycolytic activity (Table 1) [48]. Sema3A-mediated Plexin-A1/NRP-1-mediated signal transduction was also found to inhibit the proliferation of brain tumor stem cells. Surprisingly, sema3A nevertheless concomitantly promoted the invasive activity of these cells (Figure 2) (Table 1) [49].

#### 2.1.2. Plexin-A2

Plexin-A2 transduces signals of class-3 semaphorins, such as sema3B. Plexin-A2 mRNA expression decreased in mouse genetic models of tumor progression, such as the RipTag2 model [79,80]. Plexin-A2 is the main plexin receptor that binds to NRP-1 and NRP-2 and transduces sema3B signals in association with plexin-A4 [42]. Sema3B is a tumor suppressor gene. Its expression is lost in lung cancer, liver cancer, and neuroblastoma [81,82,83]. Plexin-A2 expression was upregulated in metastatic prostate cancer tumors. In contrast to the above-mentioned studies, plexin-A2 upregulation enhanced the migration and invasion of prostate cancer cells (Table 1) [50]. Plexin-A2 expression was also upregulated in glioblastomas [78]. An in-silico study identified plexin-A2 as a gene that is never mutated in malignant melanomas and may thus be essential for malignant melanoma development and tumor progression [84]. The expression of sema3C, NRP-1, and plexin-A2 is upregulated by monoamine oxidase A (MAOA) in prostate cancer cells. The produced sema3C then binds to NRP-1, which induces the formation of a signaling complex containing NRP-1, plexin-A2, and the MET tyrosine-kinase receptor. This, in turn, activates MET-mediated signal transduction and induces perineural invasion by prostate cancer cells (Table 1) (Figure 3) [51].

KIAA1199 was identified as BCL-3 and p65 expression-dependent gene in transformed keratinocytes. Its expression is enhanced following human papillomavirus (HPV) infection of keratinocytes. Interestingly, KIAA1199 also binds to the intracellular domain of plexin-A2 and inhibits sema3A-induced apoptosis of cervical cancer cells (Table 1) (Figure 3) [52]. Plexin-A2 was also found to enable the development of tumors from glioblastoma-derived cells. Knock-out of plexin-A2 in these cells inhibited their proliferation and induced senescence and cell cycle arrest. The intracellular domain of plexin-A2 was found to be essential for these pro-proliferative effects. Re-expression of plexin-A2 mutated in its semaphorin binding domain in plexin-A2 knock-out glioblastoma-derived cells did not abrogate rescue of cell proliferation, suggesting that the pro-proliferative effect of plexin-A2 is not activated by the binding of a semaphorin to plexin-A2 (Table 1) [53].

#### 2.1.3. Plexin-A3

Plexin-A3 forms complexes with NRP1 and NRP2 and transduces the repulsive signal of sema3F and sema3A [85,86,87,88]. Sema3F is an inhibitor of tumor angiogenesis and tumor lymph angiogenesis [89,90] and inhibits lymph vessels-mediated metastasis of head and neck squamous carcinoma cells [88]. Over-expression of plexin-A3 was found to inhibit cell invasion of epithelial ovarian cancer cells (Table 1) [54].

#### 2.1.4. Plexin-A4

Plexin-A4 transduces signals of class 3 semaphorins such as sema3A and 3B [26,42,44] and of class-6 semaphorins [42,91,92]. Plexin-A4 in association with plexin-A1 transduces inhibitory signals of sema3A in U87MG glioblastoma cells. In addition, plexin-A4 transduces pro-proliferative signals induced by sema6B in glioblastoma cells. The development of tumors in athymic nude mice from implanted U87MG glioblastoma cells in which plexin-A4 or sema6B expression was silenced is strongly inhibited. Likewise, silencing the expression of plexin-A4 in various lung cancer-derived cells or in malignant melanoma-derived cells also inhibited the proliferation of these cells (Table 1) (Figure 3) [26]. Additionally, in non-small cell lung carcinoma cells, inhibition of plexin-A4 expression using microRNA (miR)-564 inhibited the proliferation and migration of these cells (Table 1) [55]. Plexin-A4 was also found to function as a negative regulator of Cytotoxic T-cell migration and proliferation, independently of semaphorins. It was more abundant in circulating Cytotoxic T-cell in melanoma patients as compared with healthy volunteers [93]. Plexin-A4 (or plexin-A1) in a complex with NRP-1 can transduce sema3A attractive signals involving VEGFR1 for tumor-associated macrophages after hypoxia induction. Once tumor-associated macrophages are in the hypoxic areas of the tumor, they remain there because of plexinA4/plexin-A1 signals. As a result, tumor-associated macrophages lose their anti-tumor immunity and promote angiogenesis, tumor growth, and metastasis [44].

### 2.2. Type-B Plexins

Type-B plexins are known to participate in the promotion of angiogenesis and tumor progression. They transduce the signals of class-3, class-4, and class-5 semaphorins and of other factors, such as the ribonuclease angiogenin [65] and the androgen-regulated-trans-membrane-serine protease gene [58].

#### 2.2.1. Plexin-B1

Plexin-B1 transduces signals of sema4D (also known as CD100) in head and neck squamous cell carcinoma (HNSCC) in a xenograft mouse model [56]. The soluble cleaved form of sema4D was found to promote epithelial-mesenchymal transition (EMT) and, consequently, tumor cells metastasis via plexin-B1 (Table 1) (Figure 3) [56]. In Osteosarcoma cells, sema4D-induced plexin-B1 signaling activates the Pyk2-PI3K/AKT pathway and promotes cell invasion, proliferation, and migration (Table 1) (Figure 3) [57]. In castration-resistant prostate cancer, plexin-B1 was observed to transduce the signals of sema3C together with NRP-1/2 and promote cancer growth by activation of EGFR, ErbB2, and MET (Table 1) (Figure 3) [31]. The TMPRSS2-ERG fusion gene was found to enhance the expression of plexin-B1 in prostate cancer cells. The upregulated plexin-B1 was observed to enhance TMPRSS2-ERG induced prostate cancer-derived VCaP cell migration and invasion (Table 1) [58]. In ovarian cancer, plexin-B1 was found to be upregulated in tumor tissues of patients as compared with healthy ovaries and to be in correlation with lymphatic metastasis. Knockdown of plexin-B1 in ovarian cancer-derived cells results in inhibition of AKT phosphorylation, re-arrangement of the actin cytoskeleton, and inhibition of tumor cell migration and invasion (Table 1) [59].

Opposite results were reported in a study that examined the role of plexin-B1 in breast cancer metastasis. In that study, knockdown of plexin B1 in MDA-MB-231 cells led to a remarkable increase in the motility of breast cancer cells (Table 1) [60]. Plexin-B1 was also found to function as a tumor suppressor gene in primary melanoma harboring an activated Raf oncogene. The ability of plexin-B1 to suppress tumorigenesis depended on its GAP domain (Table 1) [61]. Additionally, plexin-B1 and plexin-B2 were also found to function as tumor suppressors in basal cell carcinoma. Inhibition of the expression of these two plexins in these cells resulted in increased cell proliferation and elevation of nuclear YAP activity. Loss of plexin-B1/plexin-B2 could represent a mechanistic basis for the loss of contact inhibition, a hallmark of cancer (Table 1) [62].

#### 2.2.2. Plexin-B2

Plexin-B2 functions as a receptor for the four class-4 semaphorins. Sema4C was found to promote the proliferation of breast cancer cells, and its expression in breast cancer is correlated with poor prognosis. Knock-down of either plexin-B2 or sema4C leads to cell cycle arrest, senescence, and cytokinesis defects in breast cancer-derived cells. Following sema4C binding, plexin-B2 was found to associate with ErbB2 and to activate PDZ-RhoGEF/LARG, an exchange factor for RhoA. Activation of RhoA was found to be critical for the proliferation and development of estrogen-independent tumor metastasis in mice (Table 1) (Figure 3) [63]. Plexin-B2 also transduces sema4C signals in glioblastoma cells. This, in turn, was found to be dependent on its GAP domain leading to the activation of RhoA and Rac1, promoting glioma and glioblastoma cell invasiveness [64,94]. Activation of plexin-B2 with sema4C also acts synergistically with HGF-activated MET to promote MET phosphorylation and to promote the progression of glioblastoma (Table 1) (Figure 3) [64].

Interestingly, plexin-B2 also functions as a receptor for angiogenin (ANG) [65], a secreted ribonuclease [95]. Plexin-B2 was found to mediate ANG stimulation of rRNA transcription, ANG-induced phosphorylation of AKT and ERK, and the resulting proliferation of tumor cells. Targeting the ANG-binding site of plexin-B2 with antibodies results in inhibition of the development of xenograft tumors in mice (Table 1) [65]. In prostate cancer, plexin-B2 plays a role together with angiogenin in the regulation of the stemness of prostate cancer stem cells (CSC). Inhibition of ANG and Plexin-B2 reduced CSC stemness and elicited CSC exhaustion. Inhibitors of angiogenin/plexin-B2 were also found to sensitize prostate CSCs to chemotherapy (Table 1) (Figure 3) [66]. Plexin-B2 was also found to promote glioblastoma progression induced by ANG/plexin-B2. Inhibition of either plexin-B2 or ANG was found to inhibit glioblastoma cell proliferation, invasion, and tumor growth in animal models (Table 1) (Figure 3) [67].

In ovarian cancer biopsies, plexin-B2 was upregulated together with the circular RNA, circ_0013958. Circ_0013958 prevents plexin-B2 downregulation by miR-637 and, as a result, promotes cell proliferation, migration, invasion, and tumor growth (Table 1) [68]. In stem cells from cancers of unknown primary tumors, plexin-B2 that was mutated in its extracellular IPT domain (G842C) is constitutively active, promoting their ability to proliferate and their tumor-forming ability. Mutated plexin-B2 also promotes the invasiveness of these stem cells via association with and activation of EGFR (Table 1) [69].

#### 2.2.3. Plexin-B3

Plexin-B3 was found to transduce inhibitory signals of sema5A in glioma cells. It was found to inhibit glioma cell invasion and migration by induction of Rac1 inactivation (Figure 2). Furthermore, plexin-B3 activated by sema5A interacts with protein kinase C, which induces disassembly of actin stress fibers and phosphorylate actin-binding protein fascin-1. Fascin-1 and glial fibrillary acidic protein, GFAP, which is upregulated by sema5A, induce morphological changes in glioma cells and astrocytic differentiation of gliomas (Table 1) [70]. Additionally, plexin-B3 knockdown was found to promote the migration and invasion of pancreatic cancer cells and to induce tumor metastasis in mouse models (Table 1) [71]. Loss of plexin-B3 expression was also in correlation with worse prognosis in hepatocellular carcinoma patients [96]. In contrast, in triple-negative breast cancer cells, plexin-B3 knockdown inhibited cancer cell growth and cell migration and invasion, as well as tumor progression in animal models (Table 1) [72].

### 2.3. Plexin-C1

Plexin-C1 functions as a receptor for sema7A [97]. It is a tumor suppressor that inhibits progression of melanomas. Plexin-C1 expression is downregulated in primary malignant melanoma as compared with melanocytic nevi and is lowest in metastatic melanoma. Sema7A binding to plexin-C1 upregulates Lim kinase II, which phosphorylates cofilin and inactivates cofilin, which inhibits tumor progression (Figure 2) (Table 1) [73]. In human glioma cells, it was observed that Growth Arrest-Specific 5 (GAS5) is a tumor suppressor which downregulates miR-222 and, as a result, upregulates plexin-C1 and inactivates cofilin, resulting in inhibition of tumor progression (Table 1) [74]. The binding of sema7A to plexin-C1 and β1 integrin receptors stimulates the expression of the Chi3l1 gene, which promotes metastasis of melanoma and breast cancer cells (Table 1) [75]. In hepatocellular carcinoma (HCC), plexin-C1 was overexpressed. The expression was correlated with the histological grade of HCC and overall survival [98,99]. Plexin-C1 was also found to be upregulated in tumors of patients with stomach adenocarcinoma as compared to non-tumor tissues. In these tumors, it was found that plexin-C1 expression is regulated by interferon regulatory factor-5 (IRF5) and that the expression is highly correlated with the presence of M2 macrophages which promote tumor progression [100,101]. In gastric cancer cells, plexin-C1 was found to be upregulated and to promote migration and proliferation of these cells through activation of epithelial to mesenchymal transition (EMT) and through induction of the expression of genes related to inflammation (Table 1) [76]. Additionally, plexin-C1 is over-expressed in hepatocellular carcinoma, and its expression is inversely correlated with overall survival [99].

### 2.4. Plexin-D1

Plexin-D1 is expressed aberrantly in many types of solid tumors in both tumor vessels and tumor cells [102,103]. Plexin-D1 functions as a receptor for sema3E. In contrast to other class-3 semaphorins, sema3E binds to plexin-D1 directly without the requirement for neuropilins [104]. Sema3C binds to NRP-1, which then associates with plexin-D1 in endothelial cells to activate plexin-D1, resulting in inhibition of angiogenesis [105]. Interestingly, sema3C can also activate plexin-D1-mediated signal transduction in the absence of neuropilins, provided that plexin-A4 is also co-expressed along with plexin-D1 [106]. Sema3C was also reported to activate signaling via an NRP-2/plexin-A1 complex in lymphatic endothelial cells and to inhibit lymphangiogenesis [107]. Plexin-D1 mediates inhibitory signals induced by full-length sema3E to inhibit tumor angiogenesis and tumor metastasis [35]. Sema3E is cleaved by furin-like pro-protein convertases, which are abundant in the microenvironment of tumors, to produce p61-Sema3E, a cleaved active variant of sema3E [108]. The p61-sema3E cleavage product is also able to bind to plexin-D1 and to inhibit angiogenesis, but in contrast to the full-length form also promotes the association of plexin-D1 with ErbB2. This results in the activation of ErbB2 that, in turn, promotes tumor cell invasiveness and tumor metastasis (Table 1) (Figure 3) [34,35]. Plexin-D1 expressed in breast cancer cells interacts, in the absence of sema3E, with the orphan nuclear receptor NR4A1 to induce apoptosis. Following the binding of sema3E, the interaction with NR4A1 is disrupted, enabling sema3E-induced tumor cell survival. However, it is not clear if this effect is mediated by full length or by the furin-cleaved p61-Sema3E form of sema3E (Figure 2) (Table 1) [77]. Plexin-D1 was also found to form complexes with plexin-A2 and NRP-1. Sema3C signaling via this complex promotes the survival of glioma stem cells via activation of Rac1 (Table 1) (Figure 3) [78]. Plexin-D1 may serve as a potential biomarker for cervical cancer, as it was found that it is strongly expressed in the endothelial cells of the cervical cancer samples, and there is no expression of plexin-D1 in endothelial cells derived from normal cervical tissues [109].

## 3. Conclusions

Plexins are a family of single-pass transmembrane receptors which have the ability to inhibit or potentiate tumor cell proliferation and invasiveness. Plexins function primarily as semaphorin receptors. Various plexins are expressed in tumor cells as well as in a variety of other cell types, which are recruited to the tumor microenvironment, such as endothelial cells. The final effects on tumor progression are, of course, an integration of all of these effects, but here we have only covered the direct effects of plexin-mediated signal transduction on the behavior of tumor cells. Interestingly, the effects that given semaphorins have on the behavior of tumor cells are strongly modulated by the association of their plexin and neuropilin receptors with additional cell surface molecules. For example, sema3A, which normally inhibits tumor cell invasiveness when signaling through its plexin-A1 receptor, induces tumor cell invasiveness following the association of its plexin-A1 receptor with cell surface-associated perlecan. It is likely that in addition to the known interactions that we have covered here, there are additional interactions that can also modulate plexin-mediated responses of tumor cells to semaphorins. The identification of such modulating interactions is likely to become the focus of future studies.

## Figures and Tables

**Figure 1 cancers-15-04046-f001:**
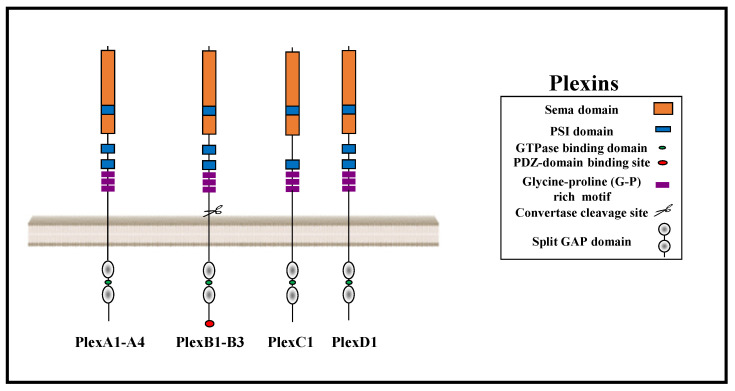
The structure of plexins: The nine vertebrate plexins are subdivided into four type A, three type B, and one each of plexins C and D. Plexins are single-pass transmembrane receptors distinguished by the presence of a split cytoplasmic GTPase-activating protein (GAP) domain that binds small GTPases such as Rho and Rac1. The extracellular domains of all plexins contain a sema domain, PSI (Plexin, Semaphorin, Integrin) motifs, and immunoglobulin-plexin-transcription (IPT) glycine–proline (G–P)-rich motifs, which the plexins share with the tyrosine kinase receptors belonging to the MET tyrosine-kinase receptors family.

**Figure 2 cancers-15-04046-f002:**
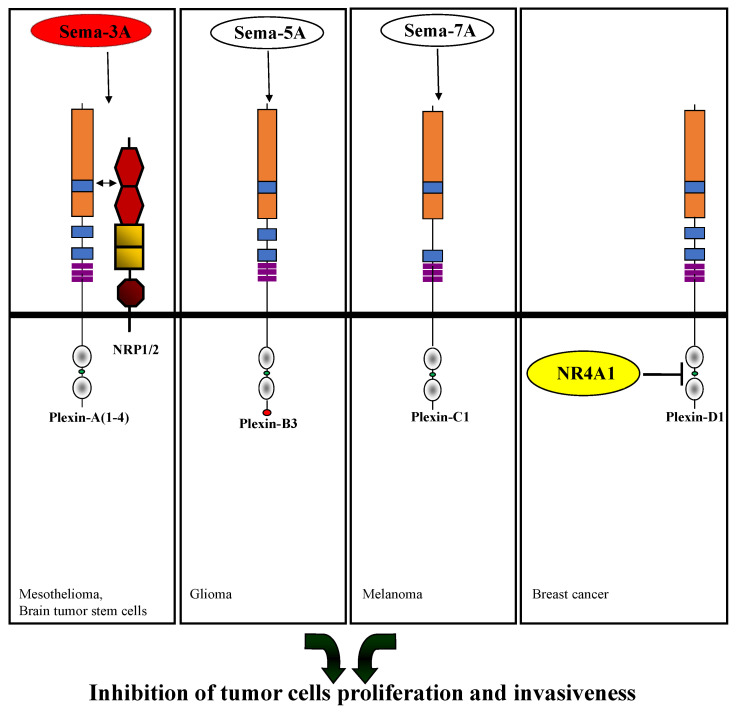
Inhibition of tumor cell proliferation and invasiveness through plexin-mediated signal transduction. Shown are the effects of semaphorins that were found to inhibit tumor cell proliferation or tumor cell invasiveness following their binding to the indicated plexins. Semaphorins that can both inhibit or promote tumor progression depending on specific interactions of their plexin receptors with additional proteins are shown on a red background. In the case of sema3A, neuropilin is required in addition to a type A plexin. The intracellular domain of plexin-D1 was found to associate with the nuclear orphan receptor NR4A1, which is depicted on a yellow background. This, in turn, induces apoptosis of the breast cancer cells. The references shown at the bottom of the panels direct to the relevant manuscripts.

**Figure 3 cancers-15-04046-f003:**
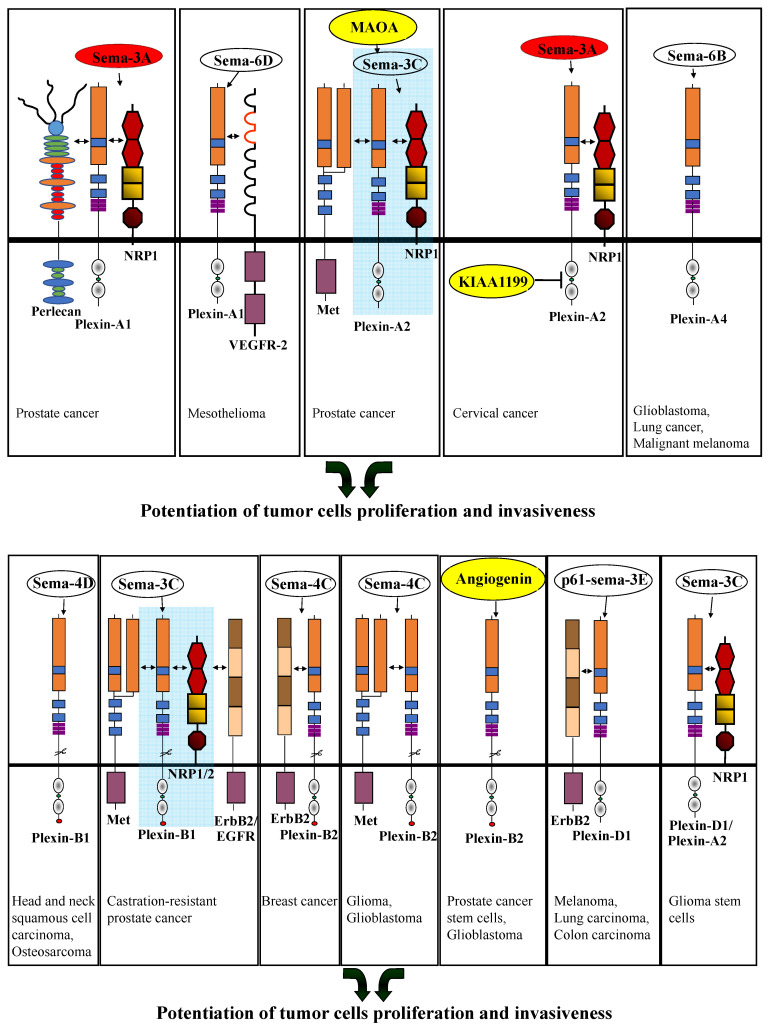
Potentiation of tumor cell proliferation and invasiveness through plexin-mediated signal transduction. Shown are the effects of semaphorins that were found to promote tumor cell proliferation or tumor cell invasiveness following their binding to the indicated plexin or neuropilin/plexin complexes in the case of the class-3 semaphorins. Semaphorins that can both inhibit or promote tumor progression depending on specific interactions of their plexin receptors with additional proteins are shown on a red background. Modulators of plexin-mediated signal transduction, which interact with the plexins, such as the cell surface receptors MET, VEGFR2, and perlecan, are depicted. Other molecules that affect signal transduction by various plexins, such as angiogenin which functions as an alternative plexin-B2 ligand, and MAOA and KIAA1199, which functions as modulators of sema3C or sema3A signaling, are depicted on a yellow background. Double-headed arrows indicate association with additional cell surface receptors, such as various tyrosine-kinase receptors, which can be activated following the binding of various semaphorins to their plexin receptors to promote tumor cell proliferation and invasiveness. The references shown at the bottom of the panels direct to the relevant manuscripts.

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
