# Peer review of "Plexins as Regulators of Cancer Cell Proliferation, Migration, and Invasivity"

_cancers, 2023, doi:10.3390/cancers15164046_

Round 1

Reviewer 1 Report

In “Plexins as Regulators of Cancer Cells Proliferation, Migration and Invasivity”, Toledano and Neufeld reviewed the effects of plexins in different types of cancer focusing on the effect of plexins on tumor cells and the triggered signal transduction upon ligands binding.

Major comments:

1.       The bibliography in this field is properly covered and well organized in general terms but I think that, since the manuscript provides detailed information about all plexins in different types of tumor cells a table summarizing the information should improve this manuscript. Two figures are provided to summarize the available information, but in that figures there is no information explaining in whick type of tumor it was identified each transduction signal. This is important since some times, depending on tumor type and cellular contexts, the effects of plexins could be opposites. The table should describe the different roles of each plexing (increasing or decreasing proliferation, migration, ect) detailing in which type of tumor cells was reported, witch ligands and coreceptors participates and the bibliographic cites.

Minnor comments:

In line 106: “VEGRF2 and with the fibroblast growth factor tyrosine-kinase receptor FGFR1.”, I think that VEGRF2 should be replaced by VEGFR2.

Line 119: “proliferation migration and invasiveness of tumor cells” should be replaced by “proliferation, migration and invasiveness of tumor cells”.

The first paragraph in line 125 should be removed.

In line 130, in the sentence “In general, activation of these plexins by class-3 semaphorins as well as by class-6 semaphorins usually inhibits tumor angiogenesis and when expressed in tumor cells these plexins also usually inhibit tumor progression. However, this is a generalization and their association with additional modulators can lead to opposite effects.”, I consider that “In general” should be removed since is redundant with “usually”.

In line 240, the sentence “. In a xenograft mouse model” should be replaced be “in a xenograft mouse model”.

In line 282, where say “Trargeting” should say “Targeting”.

In my minor comments I have detailed some writing errors. Considering that I have detected quite a few errors, I suggest that the authors reread the work to correct any errors in writing

Author Response

Replies to Reviewer 1

  1. The bibliography in this field is properly covered and well organized in general terms but I think that, since the manuscript provides detailed information about all plexins in different types of tumor cells a table summarizing the information should improve this manuscript. Two figures are provided to summarize the available information, but in that figures there is no information explaining in whick type of tumor it was identified each transduction signal. This is important since some times, depending on tumor type and cellular contexts, the effects of plexins could be opposites. The table should describe the different roles of each plexing (increasing or decreasing proliferation, migration, ect) detailing in which type of tumor cells was reported, witch ligands and coreceptors participates and the bibliographic cites.

We thank the reviewer for his helpful comment. We have now added a table (Table 1) which summerises the effects of the plexins and their various ligands and associated molecules on the behaviour of tumor cells from various types of cancers. Figures 2 and 3 show various combinations of plexins, associated molecules and their various ligands. We have now added to the various panels of these two figures information directing the readers to the relevant references.

Minnor comments:

In line 106: “VEGRF2 and with the fibroblast growth factor tyrosine-kinase receptor FGFR1.”, I think that VEGRF2 should be replaced by VEGFR2.

We changed that as requested (see line 74).

Line 119: “proliferation migration and invasiveness of tumor cells” should be replaced by “proliferation, migration and invasiveness of tumor cells”.

We changed that as requested (see line 87).

The first paragraph in line 125 should be removed.

The paragraph was removed.

In line 130, in the sentence “In general, activation of these plexins by class-3 semaphorins as well as by class-6 semaphorins usually inhibits tumor angiogenesis and when expressed in tumor cells these plexins also usually inhibit tumor progression. However, this is a generalization and their association with additional modulators can lead to opposite effects.”, I consider that “In general” should be removed since is redundant with “usually”.

"In general" was removed as per the suggestion of the reviewer (see line 96).

In line 240, the sentence “. In a xenograft mouse model” should be replaced be “in a xenograft mouse model”.

We changed that as requested (see line 217).

In line 282, where say “Trargeting” should say “Targeting”.

We changed that as requested (see line 259).

Reviewer 2 Report

Dear authors,

I read your review with great interest. I appreciated the description that was made about plexins. The authors also cited recent works. Furthermore, the authors adequately explain the role of plexins in terms of migration and invasioness.

In my opinion, I believe that the review represents a good collection of data about plexins and does not need further modifications

Author Response

This reviewer had no comments.

Author Response

This reviewer apparently refers to a manuscript different than ours. We do not understand how this pertains to our manuscript